# Molecular Imaging and Quantification of Smooth Muscle Cell and Aortic Tissue Calcification In Vitro and Ex Vivo with a Fluorescent Hydroxyapatite-Specific Probe

**DOI:** 10.3390/biomedicines10092271

**Published:** 2022-09-14

**Authors:** Anna Greco, Jaqueline Herrmann, Milen Babic, Manasa Reddy Gummi, Markus van der Giet, Markus Tölle, Mirjam Schuchardt

**Affiliations:** Department of Nephrology and Medical Intensive Care, Corporate Member of Freie Universität Berlin and Humboldt Universität zu Berlin, Charité—Universitätsmedizin Berlin, Hindenburgdamm 30, 12203 Berlin, Germany

**Keywords:** fluorescence staining, mineralization, quantification, vascular smooth muscle cell

## Abstract

Vessel calcification is characterized by the precipitation of hydroxyapatite (HAP) in the vasculature. Currently, no causal therapy exists to reduce or prevent vessel calcification. Studying the underlying pathways within vascular smooth muscle cells and testing pharmacological intervention is a major challenge in the vascular research field. This study aims to establish a rapid and efficient working protocol for specific HAP detection in cells and tissue using the synthetic bisphosphonate fluorescence dye OsteoSense™. This protocol facilitates especially early quantification of the fluorescence signal and permits co-staining with other markers of interest, enabling smaller experimental set-ups with lesser primary cells consumption and fast workflows. The fluorescence-based detection of vascular calcification with OsteoSense™ combines a high specificity with improved sensitivity. Therefore, this methodology can improve research of the pathogenesis of vascular calcification, especially for testing the therapeutic benefit of inhibitors in the case of in vitro and ex vivo settings.

## 1. Introduction

Arteriosclerosis is a pathophysiological disorder resulting from an abnormal synthesis of hydroxyapatite crystals (HAP) by vascular smooth muscle cells (VSMC) and represents a strong predictor of cardiovascular disease (CVD) [1]. Many factors, especially uremia, are associated with an increased prevalence of medial arterial calcification (MAC) that declines the life expectancy of patients [2]. Therefore, the current effort in that research field is to identify signaling pathways and targets suitable for a therapeutic approach [3]. Several in vivo, in vitro and ex vivo experimental settings and various analyzing tools are available, having been recently summarized [4,5]. Here, HAP detection within the vasculature and in cell culture is crucial. To date, several assays are employed as useful tools to study vascular calcium-rich deposits: the tissue demineralization with hydrochloric acid (HCl) and subsequent quantification of the calcium amount as well as histological staining using von Kossa and Alizarin Red. However, none of these techniques are specific to HAP. Demineralization with HCl extracts free calcium ions subsequently quantified via a photometric assay. Similarly, anthraquinone derivate Alizarin Red stains calcium, while von Kossa stains phosphates. Thus, selective and specific HAP staining might improve the detection of vascular calcification, especially regarding the screening of therapeutic drugs in the in vitro, ex vivo and in vivo models [5]. The advantages of specific HAP-binding and detection via a fluorescence-based methods were already proven [6].

This study aims to bring specific HAP staining with an available and easy-to-use dye, OsteoSense™680EX, in in vitro and ex vivo protocols. To assess the correlation between the OsteoSense™680EX staining protocols and existing, broadly applied methods for detection and quantification of vascular calcification, we compared the results generated with OsteoSense™680EX to Alizarin Red staining and photometric quantification. OsteoSense™680EX was used for HAP staining in a time-response curve in vitro, combined with subsequent quantification of fluorescence intensity. The functionality of the protocol was also proven for co-staining with a biomarker of DNA-double strand breaks, gamma-histone H2A.X (γH2A.X). In addition, the staining works in fresh tissue and in fixed paraffin-embedded aortic tissue slides ex vivo. The protocols allow rapid and specific HAP staining from in vitro to ex vivo settings, facilitating the study of vascular mineralization with high sensitivity and an easy workflow.

## 2. Results

Identification and quantification of calcium phosphate crystals are used for several calcification models, whereby Alizarin Red staining, von Kossa staining and calcium quantification upon decalcification are the current standards [5]. However, not only the sensitivity of the current gold standards for the detection of calcification, but also their specificity is different. While Alizarin Red and the photometric assay detect calcium, von Kossa staining detects phosphate. As none of them specifically stains and quantifies HAP, here the fluorescence dye OsteoSense™ was applied to ex vivo and in vitro settings.

### 2.1. Ex Vivo Staining of the Whole Thoracic Aorta

Current protocols for whole aortic tissue staining ex vivo mainly use Alizarin Red [5,7]. For in vivo staining, OsteoSense™ was already successfully used [8]. Even though to our knowledge, fluorescence dyes for detection of calcification were never tested for ex vivo application, we expected ex vivo staining of calcification crystals with fluorescence dyes to work. We tested OsteoSense™ for whole aortic staining upon 7 d of incubation with control medium (COM) and calcification medium (CAM) ex vivo. As shown in Figure 1, several HAP crystals could be detected in CAM incubated aortic tissue, whereas no fluorescence signal is detectable in COM stimulated tissue. In addition to visualization of the localization of HAP in the whole thoracic aorta, the fluorescence signal could also be directly quantified and significantly increased in CAM incubated aorta compared to COM treated material (Figure 1B). Afterwards, the calcium content of the OsteoSense™ treated tissue was also quantified via conventional decalcification of the same tissue and subsequent photometric calcium assay (Figure 1C), revealing good correlation between both protocols.

### 2.2. Ex Vivo Staining of Mineralization in Aortic Tissue Slides

The ex vivo stimulation of aortic rings is, in addition to in vitro stimulation of VSMC and in vivo animal models, a standard protocol to study the pathogenesis of vascular calcification [4]. Usually, the calcification is visualized by Alizarin Red or von Kossa staining and quantification of the calcium content is carried out upon decalcification of the tissue in HCl [5]. Here, we used OsteoSense™ to stain HAP upon 7 d and 14 d of incubation in CAM and respective control (COM). We compared OsteoSense™ staining to both Alizarin Red staining from paraffin-embedded aortic tissue slides and photometric calcium quantification upon tissue decalcification. The staining methods confirmed the absence of HAP and calcium in COM stimulated rings upon 7 and 14 d. In CAM stimulated aortic tissue a medial located HAP and calcium content could be visualized. For both staining methods, the location of the respective staining matches in the serially cut rings (Figure 2A for 7 d, Appendix A for 14 d). As the signal-to-noise ratio is very good for the fluorescence staining, the signal could be easily quantified and normalized to the nuclei areas. Here, a significant increase in HAP content was found for the CAM stimulated rings compared to their respective controls (Figure 2B). This result is comparable with the current standard of calcium quantification with a photometric calcium assay after decalcification of the tissue (Figure 2C).

### 2.3. In Vitro Staining of Mineralization

In literature, the stimulation time to achieve robust quantifiable calcification of VSMC in vitro varies from 3 to 21 days [4]. Reasons for these discrepancies in incubation might be the use of cells of different origins (human, rat, mice) and different calcification inducers [4]. In addition, the sensitivity of the current gold standard methods for the detection of calcification are different. With the benefit of specificity and the increase in sensitivity by fluorescence quantification, the objective was to reduce the required treatment time for robust calcification detection below the most common treatment periods between 7 and 14 d [4,9,10]. A reduction of the timeframe required for robust detection of calcification is reasonable to reduce material consumption as well as work expenditure and facilitate high-throughput analysis, especially for inhibitor screening experiments. We found a specific HAP staining as early as after 24 h of stimulation with CAM that time-dependently increased over 48 h, 72 h, and 7 d (Figure 3A).

Fluorescence signal quantification confirmed a significant and time-dependent increase in HAP upon stimulation with CAM. The results were also confirmed by photometric calcium quantification (Figure 3C). Even though the results obtained with fluorescence imaging and photometric calcium assays are comparable, we detected a higher discrimination regarding the detected effect sizes with the fluorescence staining.

### 2.4. In Vitro Co-Staining of Mineralization and DNA Damage

From current literature, there is evidence of DNA damage in the vascular calcification process [11,12]. To further verify this aspect on a single cell level, we aimed to multiplex the calcification detection of OsteoSense™680EX with fluorescence-based detection of DNA damage via γH2A.X. Multiplexing on a single cell level not only allows reduction of primary cells in the thought of the ThreeR (3R), referring to the guiding principles “Replace, Reduce, Refine” for more ethical use of animals for scientific research formulated by Russel and Burch [13], but also permits a direct comparison of the markers of interest in the signaling cascade in one cell [14]. As already known from our own and others’ previous studies, doxorubicin is a potent inducer of DNA damage and calcification in VSMC and was therefore used as a positive control [12,14,15], in addition to the stimulation of cells with COM and CAM as before.

As given in Figure 4, the current protocol allows co-staining of HAP and γH2A.X.

## 3. Discussion

Here, we demonstrate that the HAP-specific fluorescent dye OsteoSense™ sensitively stains micro-calcifications in vitro in rat VSMC, ex vivo in whole rat aortas and in rat aortic ring sections. We expect the fluorescence-based detection of HAP to offer advantages over the current gold standards for ex vivo and in vitro staining in the research field of vascular calcification.

The advantages of specific HAP-binding and detection via a fluorescence-based method, using a fluorescein-bisphosphonate probe, were already shown by Sim et al. [6]. Vascular calcification was identified in in vitro models and aortic tissue as well as tissue sections with higher sensitivity and specificity than with the current gold standards [6]. Yet as a disadvantage, the fluorescein-bisphosphate needs synthesis and time-consuming purification by reverse-phase high-pressure liquid chromatography by the user [6]. With OsteoSense^TM^, a commercially ready-to-use fluorescent dye coupled to a bisphosphonate is available that binds to HAP crystals with exquisite affinity [5,6]. It enables easy assessment of ectopic micro-calcification sites and subtle osteogenic activities within the vasculature. The fluorescent dye was, e.g., used to detect calciprotein particles in serum [16]. The usage of OsteoSense^TM^ has also widely improved the visualization of calcium crystals in in vivo animal models and for organoid culture [8]. Yet while this HAP-specific dye might offer advantages for visualization, distinction of localization and simultaneous quantification of vascular HAP formation, to our knowledge, OsteoSense^TM^ application in vitro and ex vivo has not yet been systematically compared to the current gold standards of histological staining and photometric calcium quantification, which are so far more widely used in vascular calcification research. Here, we use the commercially available OsteoSense^TM^ fluorescence dye to transfer its advantage of specific HAP staining towards ex vivo and in vitro settings. 

The current gold standards for detection of calcification deposits in vessels or in vitro are photometric quantification of the calcium content upon demineralization of tissue or cells and histological staining via Alizarin Red and von Kossa [5]. Even though these methods offer the advantage of easy standardizable use and their applicability for tissue and cells, one of their main disadvantages is their non-specificity against HAP. While the photometric assay and Alizarin Red detect calcium, von Kossa stains for phosphate, whose presence is ubiquitous [6,17,18]. In addition, Alizarin Red also stains glycosaminoglycan or proteins involved in cell signaling pathways that will mistakenly contribute to the staining and may also interfere with magnesium, manganese, barium, strontium, and iron [6,17]. Although the error of non-specific staining seems to be low, the non-specificity of the traditional staining methods could be overcome with HAP-specific fluorescence dyes, such as fluorescein-bisphosphonate [6] and OsteoSense™ [8,16,19,20] applicable in vitro and ex vivo. Furthermore, while demineralization permits extraction of calcium and rapid quantification, it does not allow visualization of mineralization nodules, no distinction in localization (intimal vs. medial), and is destructive to the sample material [5]. In comparison, the fluorescent OsteoSense^TM^ signal is easily quantifiable and combines assessment of localization with the extent of HAP formation by dissipating a reduced amount of sample material. This advantage is of remarkable importance especially in the light of the ethical efforts to reduce the number of animals to a feasible minimum. 

Currently, for the whole tissue experimental setting, the most common staining method for calcium crystals is Alizarin Red [7,21,22]. Therefore, we first tested the applicability of OsteoSense^TM^ for whole aortic HAP staining. We used a setting of ex vivo incubation of the whole aorta in calcification inducing medium for 7 d and stained afterwards with OsteoSense^TM^680EX. It nicely visualizes HAP nodules over the whole tissue. The fluorescence signal could be easily quantified, and decalcification of the tissue afterwards is possible. In addition, for the histological analysis the fluorescence-based HAP staining has advantages regarding specificity, sensitivity and quantifiability in comparison to Alizarin Red.

One of the main objectives was also to establish and assess the applicability of in vitro HAP staining of VSMC. A similar proceeding was already used for staining of calcified human VSMC upon three weeks of stimulation [23]. However, the main advantage of the application of OsteoSense™ is the staining’s sensitivity, allowing a severe reduction of the incubation time. The assessment of HAP formation with OsteoSense™ offered the opportunity to reduce the growth area and corresponding cell number (in comparison to a photometric calcium assay) and enabled a high-throughput possibility per slide, which is particularly important for comparability of inhibitor studies. In addition, co-staining of several markers, e.g., DNA damage via γH2A.X, allows single-cell-based visualization and direct comparison of, e.g., a senescence or osteogenic marker in a single cell with HAP crystals surrounding the cell [14]. For induction of DNA damage, the known inducer doxorubicin was used [12,14,15]. The combination of OsteoSense™ detection with other fluorescence-labeled senescence or osteogenic markers is also possible and therefore offers advantages concerning a reduced primary cell consumption according to the 3R principles [13] and additional experimental read out information. In addition, the OsteoSense™ staining method could be transferred to cells or tissue from other species and thus opens a wide range of applications in the research field of vascular calcification with all the described advantages over the current gold standards.

The OsteoSense™ reagent costs are higher than the fluorescein-bisphosphonate probe and the canonical histological staining methods such as Alizarin Red and von Kossa, yet its commercial availability provides a ready-to-use probe which reduces experimental complexity. The labor efforts are diminished by a lesser sample consumption and a simple, robust, and fast workflow.

In conclusion, the application of the fluorescence-based HAP staining could not only improve the visualization in animal models, but also enhance the specific and sensitive HAP visualization and quantification in ex vivo and in vitro settings. The direct visualization and quantification in one experimental setting and especially the possibility of co-staining with other fluorescence markers increases comparability and co-localization and reduces animal material in the prospect of 3R [13]. 

## 4. Materials and Methods

### 4.1. Animals

Male Wistar rats were purchased from Janvier Labs (Le Genest-Saint-Isle, France). Euthanasia of animals was accomplished with an intraperitoneal injection of sodium pentobarbital (rats: 400 mg/kg body weight).

### 4.2. Preparation of Aortic Tissue

For the whole aorta and aortic ring staining experiments, the thoracic aorta, including the aortic arch, was isolated from Wistar rats (mean age 2–3 months, male). After euthanasia, the aorta was dissected, and the adventitia removed prior to a washing step in PBS supplemented with antibiotics.

For ex vivo whole thoracic aortic stimulation, the dissected aorta was placed in a 60 mm dish pre-filled with CAM or COM and incubated for 7 d at 37 °C and 5% carbon dioxide (CO_2_). Medium was changed every 2–3 d. For ex vivo aortic ring stimulation, the dissected aorta was sectioned in aortic rings of equal size and incubated in a well-plate with the respective stimulation medium for 7 d or 14 d at 37 °C and 5% CO_2_ with medium changes every 2–3 d. Each stimulation contained several aortic rings from different aortic parts that were equally distributed between stimulation and respective controls. After stimulation, aortic rings were either fixed overnight in 4% buffered formaldehyde, transferred to 70% ethanol, and embedded in paraffin via automatic procedure, serially cut into 4 μm sections and stained with the desired procedure or utilized for photometric calcium quantification. 

### 4.3. Culturing of Primary Vascular Smooth Muscle Cells

All cell culture components were obtained from Biochrom AG (Berlin, Germany) and Bio&Sell (Feucht, Germany). Primary VSMC from aortic tissue (aortic arch and the thoracic aorta) of Wistar rats (mean age 1.5 months, male) were cultured by the outgrowth technique described previously [24]. VSMC grown up to passages four to six were used for experiments. Cells were cultured in a humidified incubator at 37 °C with 5% CO_2_. If not stated otherwise, VSMC were cultured in Dulbecco Modified Eagle Medium (DMEM) containing 1 g/L glucose, supplemented with 10% fetal calf serum (FCS), penicillin (100 U/mL) and streptomycin (0.1 mg/mL). Calcification was induced by exposing VSMC or aortic rings to DMEM containing 4.5 g/L glucose, supplemented with 15% FCS, 284 μmol/L ascorbic acid and 5 mmol/L inorganic phosphate, penicillin (100 U/mL) and streptomycin (0.1 mg/mL) (CAM). As COM served DMEM containing 4.5 g/L glucose, supplemented with 15% FCS and antibiotics. Calcification was induced over 1 to 14 d of stimulation with CAM and its respective control COM.

### 4.4. Photometric Calcium Quantification

For quantification of calcium content in the ex vivo experimental settings, aortic rings were decalcified in HCl for 24 h at 37 °C. Afterwards, the tissue was removed from the solution, air-dried at 37 °C and the assessed dry weight was used for normalization. The remaining HCl solutions was evaporated. Pellets obtained through this procedure were dissolved in HCl and used as input samples for the colorimetric Calcium C-test (Calcium assay kit from Sciencell^TM^, Carlsbad, CA, USA). Detection was performed via Multiskan spectrum plate reader (ThermoFisher Scientific, Dreieich, Germany). 

### 4.5. Histology and Fluorescence Staining of HAP

For specific HAP staining, the fluorescence dye OsteoSense™680EX (PerkinElmer^®^, Rodgau, Germany) was used for in vitro and ex vivo staining procedures. 

#### 4.5.1. Staining Procedure and Analysis: Aortic Tissue

After 7 d of stimulation in either COM or CAM, the whole thoracic aorta underwent fixation in a 4% Formalin/PBS solution. Following that, the aorta was washed three times in PBS and stained for 24 h at 37 °C with a solution of OsteoSense™680EX (10 pmol/L in PBS). Next, the staining solution was discarded, the tissue was washed in PBS and calcification was assessed using a ChemiDoc MP Imaging System (Bio-Rad, Feldkirchen, Germany). Quantification was performed via ImageJ.

#### 4.5.2. Staining Procedures and Analysis: Aortic Tissue Slides

Paraffin-embedded aortic tissue slides were deparaffinized with Roti-Histol^®^ (ThermoFisher Scientific, Schwerte, Germany) for 20 min at room temperature (RT). Aortic sections were hydrated by immersion into ethanol solutions (95%, 80%, 70%) for 5 min each. After a washing step into deionized water for 2–3 min, the aortic tissue slides were air-dried. A standard protocol was used for Alizarin Red (pH 4) staining. Staining with OsteoSense™680EX (100 pmol/L in PBS) was performed at 37 °C for 24 h, followed by three washing steps in PBS and counter-staining of the nuclei with Hoechst staining, according to the manufacturer’s protocol. Aortic sections were finally embedded with ProLongTM Gold antifade mount (ThermoFisher Scientific, Schwerte, Germany) and either promptly imaged or stored in the dark at 4 °C until microscopic detection via Axiovert 200M microscope (Zeiss, Jena, Germany) with Zen2 software (Blue edition, Zeiss). OsteoSense™680EX signal was quantified with Zen2 software (Blue edition, Zeiss) using a 3-Sigma-threshold approach. OsteoSense™680EX signal (Gray^2^) was normalized to the cell core area (µm^2^).

#### 4.5.3. Staining Procedures and Analysis: In Vitro

VSMC were seeded in µ-Slides slides (Ibidi, Gräfelfing, Germany) and stimulated for the desired time. 24 h prior to the end of stimulation, OsteoSense™680EX was added to the respective medium at a concentration of 100 pmol/L. Cells were then deprived of medium and washed multiple times with phosphate-buffered saline (PBS). Afterwards, cells were fixed with 4% formalin/PBS solution for 10 min at RT; to remove formalin residuals, an additional washing step was performed. Nuclei were stained with Hoechst33342 (Thermo Fisher) according to the manufacturer’s protocol. 

The co-staining of HAP with OsteoSense™680EX and DNA damage foci γH2A.X was conducted according to a previously published protocol [15]. Briefly, VSMC were seeded in µ-Slides slides (Ibidi) and stimulated for 7 d. After stimulation, cells were washed with PBS, permeabilized with TritonX/PBS (0.1%), stained with an anti-γH2A.X antibody (sc-101696) 1:500 in 10% RotiBlock/PBS for 1 h at RT, followed by incubation with an Alexa Fluor 555-coupled secondary antibody (Invitrogen, A-21429) 1:1000 in 10% Roti-immunoBlock/PBS for 1 h at RT. Afterwards, cells were washed and stained for 24 h at 37 °C with 100 pmol/L of an OsteoSense™/PBS solution. Cells were washed multiple times with PBS and nuclei were counter-stained with Hoechst33342 (ThermoFisher Scientific) according to the manufacturer’s protocol. 

All experiments were imaged under an Axiovert 200M microscope (Zeiss) with Zen2 software (Blue edition, Zeiss). The fluorescence signal was quantified with Zen2 software (Blue edition, Zeiss) using a 3-Sigma-threshold approach. The cell core area was determined with Zen2 software with manual threshold determination. OsteoSense™680EX signal (Gray^2^) was normalized to the cell core area (µm^2^).

### 4.6. Statistical Analysis

All experiments were repeated at least three times in independent experiments. Images show representative pictures. Data are presented as mean ± standard error of mean (SEM) unless indicated otherwise. All statistical analysis were performed using GraphPad Prism (version 6). Comparisons between groups were carried out by ANOVA and non-parametric test. *p* values < 0.05 were considered significant.

## Figures and Tables

**Figure 1 biomedicines-10-02271-f001:**
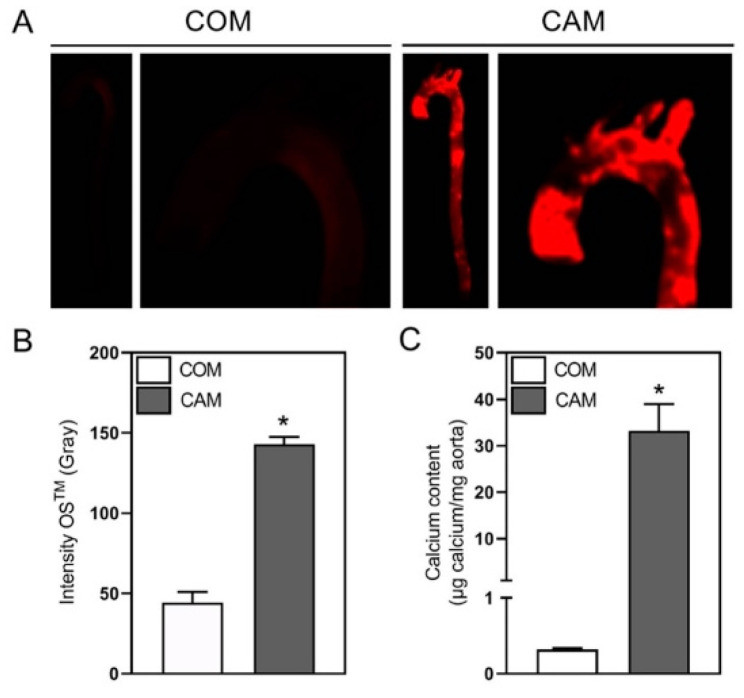
Hydroxyapatite and calcium quantification in aortic tissue. (**A**) Representative images of ex vivo thoracic aortic staining with OsteoSense™680EX (OS^TM^) upon stimulation with control medium (COM) or calcification medium (CAM), respectively, for 7 days. (**B**) Quantification of fluorescence signal. (**C**) Quantification of aortic calcium amount via photometric assay upon decalcification in hydrochloric acid. N = 3, mean ± SEM, * *p* < 0.05.

**Figure 2 biomedicines-10-02271-f002:**
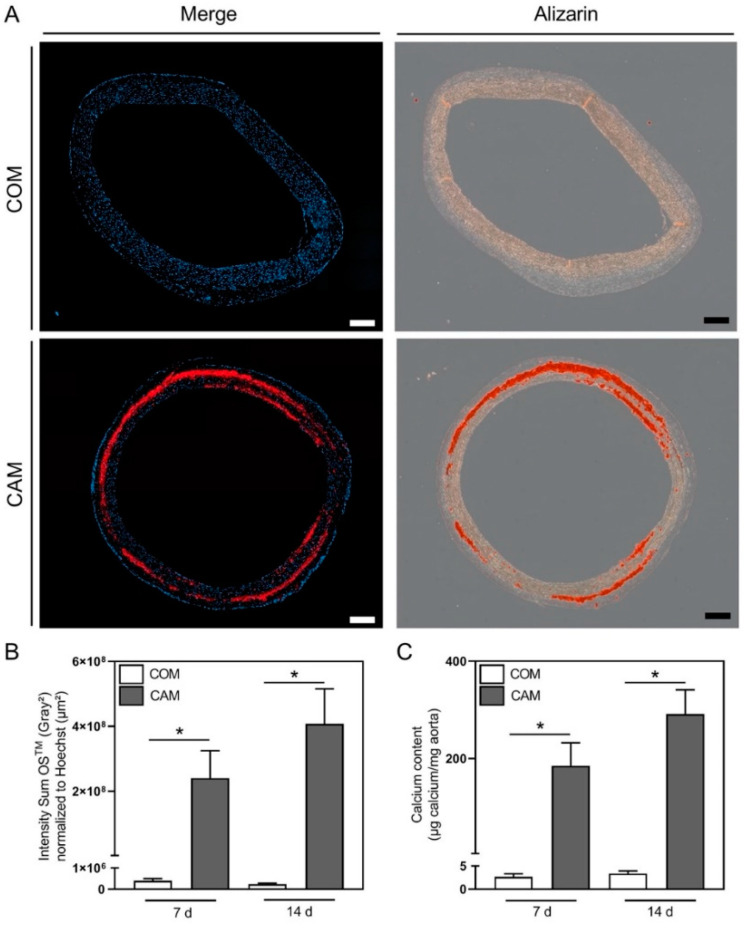
Hydroxyapatite and calcium staining in aortic tissue slides. (**A**) Representative images of aortic staining for hydroxyapatite with OsteoSense™680EX (OS^TM^) (red channel) and cell cores with Hoechst33432 (blue channel) or Alizarin Red upon stimulation with control medium (COM) or calcification medium (CAM), respectively, for 7 d. (**B**) Quantification of the fluorescence signal of OS^TM^ upon stimulation with COM or CAM, respectively, for 7 and 14 d. (**C**) Photometric quantification of aortic calcium amount upon decalcification in hydrochloric acid upon stimulation with COM or CAM, respectively, for 7 and 14 d. The scale bar indicates a 200 µm section, n ≥ 3, mean ± SEM, * *p* < 0.05 compared to marked control.

**Figure 3 biomedicines-10-02271-f003:**
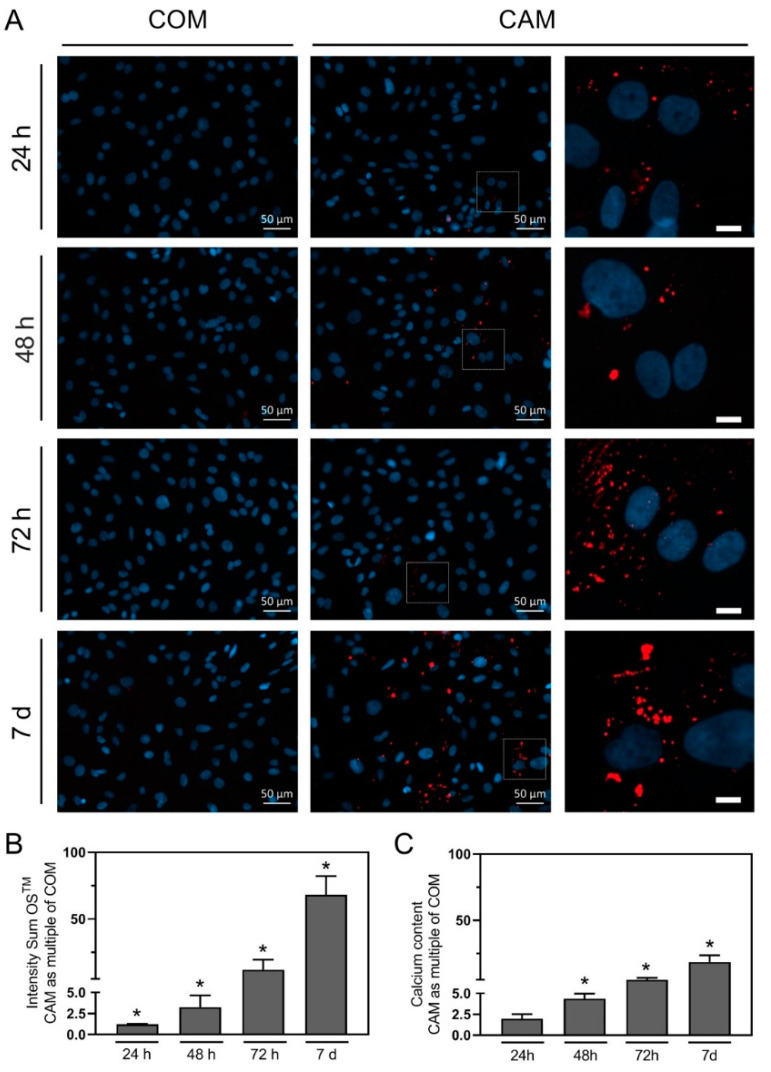
Hydroxyapatite staining and calcium quantification in vitro. (**A**) Representative images (merge) of rat vascular smooth muscle cells (VSMC) stained for hydroxyapatite with OsteoSense™680EX (OS^TM^) (red channel) and cell cores with Hoechst33432 (blue channel), after stimulation with control medium (COM) or calcification medium (CAM) for 24 h, 48 h, 72 h and 7 d. The squares indicate areas of magnifications shown on the right side. (**B**) Quantification of hydroxyapatite via fluorescence signal, VSMC were stimulated for the indicated time periods with COM and CAM. (**C**) Quantification of calcification via photometric calcium assay, VSMC were stimulated for the indicated time periods with COM and CAM. If not stated otherwise, the scale bar indicates a 10 µm section, n ≥ 3, mean ± SEM, * *p* < 0.05 in comparison to stimulation with COM.

**Figure 4 biomedicines-10-02271-f004:**
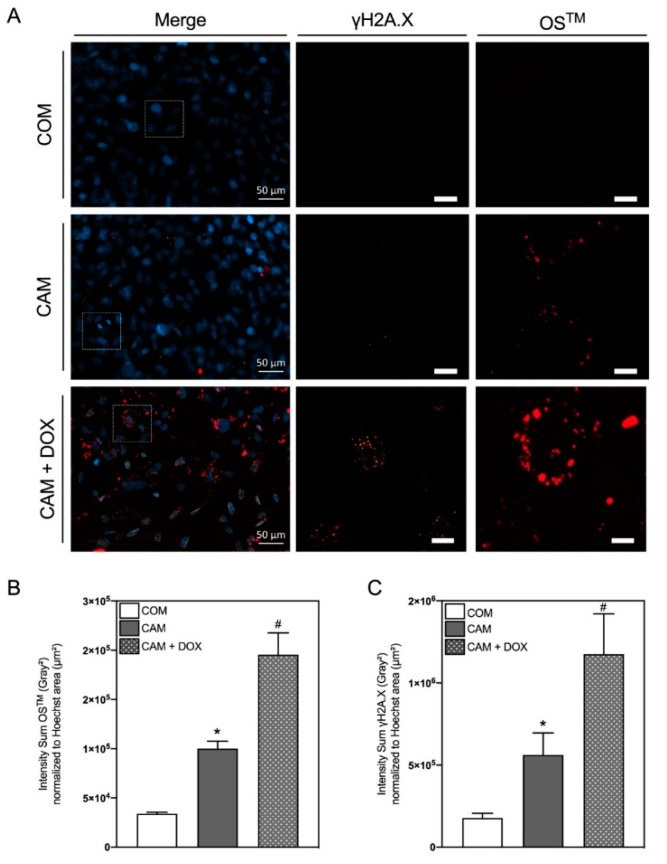
Co-staining of hydroxyapatite and gamma-histone H2A.X in vitro. (**A**) Representative images of rat vascular smooth muscle cells (VSMC) stained for gamma-histone H2A.X (γH2A.X, orange channel), cell cores (Hoechst33342, blue channel) and hydroxyapatite (HAP) (OsteoSense™680EX (OS^TM^), red channel) after stimulation in control medium (COM) or calcification medium (CAM) w/wo Doxorubicin (DOX) (500 nmol/L) for 7 d. The squares indicate areas of magnifications shown on the right side. (**B**) Quantification of OS^TM^ fluorescence signals. (**C**) Quantification of γH2A.X fluorescence signals. If not stated otherwise, the scale bar indicates a 10 µm section, n ≥ 3, mean ± SEM, * *p* < 0.05 in comparison to COM, # *p* < 0.05 in comparison to CAM.

## Data Availability

The data included in this article are available from the corresponding author upon reasonable request.

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
