# Peer review of "Molecular Imaging and Quantification of Smooth Muscle Cell and Aortic Tissue Calcification In Vitro and Ex Vivo with a Fluorescent Hydroxyapatite-Specific Probe"

_biomedicines, 2022, doi:10.3390/biomedicines10092271_

Round 1

Reviewer 1 Report

I read this paper and I think that:

The title “Molecular imaging and quantification of smooth muscle cell and aortic tissue calcification in vitro and ex vivo with a fluorescent hydroxyapatite-specific probe” is a very interesting. And this topic is essential for vascular calcification research. In this study, the authors described a protocol to study vascular mineralization with high sensitivity. 

However, there are few new findings in this study.

The fluorescence dye OsteoSense™ used in this study was not newly developed but was previously used. Also, in this study, a new research purpose or methods using OsteoSense™ dye was not suggested.

Therefore, it is likely that the authors should present data that may explain the novel advantages over existing methods to study vascular calcification.

Author Response

Thanks!

Reviewer 2 Report

1) Is there another kind of control for your experiments? If yes, please elaborate.

 2) In discussion part, I missed a summary comparison of new method compare to others.

3) Page 6, line 150: please explain “3R” abbreviation.

4) Please use homogeneous units of measurements through out your manuscript, e.g.: page 9, line 245: g/l instead of g/L; line 248: mmol/l instead of mmol/L.

 5) Figure 1: part A with COM has a very pale contour. Is there any way to observe your work more distinctive?

Author Response

Thanks!

Round 2

Reviewer 1 Report

I read revised version of manuscript.

The authors point out more clearly the advantages of the novel method in revised manuscript. This study is an interesting, and it is considered a necessary study for detection and quantification of vascular calcification.